# Proneurotrophin-3 promotes cell cycle withdrawal of developing cerebellar granule cell progenitors via the p75 neurotrophin receptor

Juan Pablo Zanin[1]*, Elizabeth Abercrombie[2], Wilma J Friedman[1]*

[1]Department of Biological Sciences, Rutgers University, Newark, United States; [2]Center for Molecular and Behavioral Neuroscience, Rutgers University, Newark, United States

**Abstract** Cerebellar granule cell progenitors (GCP) proliferate extensively in the external granule layer (EGL) of the developing cerebellum prior to differentiating and migrating. Mechanisms that regulate the appropriate timing of cell cycle withdrawal of these neuronal progenitors during brain development are not well defined. The p75 neurotrophin receptor (p75$^{NTR}$) is highly expressed in the proliferating GCPs, but is downregulated once the cells leave the cell cycle. This receptor has primarily been characterized as a death receptor for its ability to induce neuronal apoptosis following injury. Here we demonstrate a novel function for p75$^{NTR}$ in regulating proper cell cycle exit of neuronal progenitors in the developing rat and mouse EGL, which is stimulated by proNT3. In the absence of p75$^{NTR}$, GCPs continue to proliferate beyond their normal period, resulting in a larger cerebellum that persists into adulthood, with consequent motor deficits.

*For correspondence: juanpablo. zanin@rutgers.edu (JPZ); wilmaf@ andromeda.rutgers.edu (WJF)

**Competing interests:** The authors declare that no competing interests exist.

## Introduction

During development of the central nervous system (CNS) proliferation, migration and differentiation of neuronal progenitors, and the precise transition among these processes, is critical for normal development. In the early stages of cerebellar formation, granule cell progenitors (GCPs) originate from the rhombic lip of the fourth ventricle and migrate to the anlage of the developing cerebellum forming the external granule layer (EGL) (*Altman and Bayer, 1978*). In this layer, the progenitor cells proliferate extensively, and subsequently withdraw from the cell cycle and migrate toward the inner granule layer (IGL) to form the adult structure of the cerebellum. The precise control of the transition from proliferation to differentiation is key for regulating the final size of the cerebellum. The expansion of granule cell progenitors in the EGL is largely driven by sonic hedgehog (Shh) (*Dahmane et al., 1999*; *Wallace, 1999*; *Wechsler-Reya and Scott, 1999*). Disturbances in hedgehog signaling can lead to medulloblastoma, the most abundant type of pediatric tumor (*Goodrich et al., 1997*; *Zhao et al., 2015*). Several mitogenic ligands for GCPs have been identified in addition to sonic hedgehog, including insulin-like growth factor 2 (IGF2) (*Hartmann et al., 2005*) and Notch2 (*Hartmann et al., 2005*), however, little is known about factors that signal withdrawal from the cell cycle and initiation of differentiation. Since GCPs start migrating to the IGL as early as postnatal day (P) 4 and continue until P17-P20, this is a progressive process with waves of cells that exit the cell cycle and start to migrate, while others stay in the EGL and continue to proliferate (*Hatten et al., 1997*), raising the question of what regulates the withdrawal of these progenitors from the cell cycle in the continued presence of Shh.

**eLife digest** Many proteins control how cells are organised in your body. For example, a group of proteins called the neurotrophins help to control the life and death of brain cells. After a brain injury, neurotrophins can cause nerve cells to die by working together with another protein called p75NTR.

The p75NTR protein is also found in another type of brain cell called granule progenitor cells. These cells exist before birth and divide to make new cells that form a part of the brain called the cerebellum, which controls how your body moves. Granule progenitor cells do not typically die, so it is not known what p75NTR does in these cells.

Zanin et al. investigated the role that p75NTR plays in the formation of the cerebellum in mice. The experiments showed that p75NTR controls when granule progenitor cells stop producing new brain cells. The progenitor cells of mutant mice that lack this protein produced too many new cells, which resulted in these mice having larger cerebellums and being less able to control their movements. Further experiments showed that p75NTR interacts with a neurotrophin called proNT3 in the cerebellum, which is able to stop granule progenitor cells dividing even in the presence of other proteins that encourage cells to divide. An important challenge for the future is to work out why the mice lacking p75NTR are less able to control their movements.

Neurotrophins are a family of growth factors, comprised of NGF, BDNF, NT-3 and NT-4, that regulate many aspects of neuronal development and function. Neurotrophins are synthesized as precursor proteins of approximately 35 kDa that can either be cleaved to generate the mature factor, or secreted as the proneurotrophin (*Lee et al., 2001*). Mature neurotrophins bind preferentially to Trk receptors to regulate neuronal survival and differentiation, while proneurotrophins preferentially interact with the p75 neurotrophin receptor (p75NTR) in a complex with a member of the sortilin family (*Nykjaer et al., 2004*). In the cerebellum, BDNF has been shown to regulate survival and migration of GCPs via the TrkB receptor (*Schwartz et al., 1997*; *Borghesani et al., 2002*). Consistent with this role, TrkB is highly expressed throughout the EGL and IGL. In contrast, the p75NTR receptor is highly expressed in the EGL, but not the IGL (*Carter et al., 2003*). The p75NTR can mediate many different functions depending on the cell context by binding to distinct co-receptors and recruiting specific intracellular binding proteins (*Barker, 2004*; *Charalampopoulos et al., 2012*). The most well-characterized function for p75NTR is in promoting neuronal apoptosis following brain injury (*Friedman, 2010*; *Ibáñez and Simi, 2012*).

In the present work, we demonstrate a novel role for p75NTR in regulating proliferation of neuronal progenitor cells in the cerebellum. As with other cellular functions, the regulation of proliferation by p75NTR is dependent on cellular context, this receptor has been shown to promote cell cycle entry in developing retinal and cortical neurons (*Morillo et al., 2012*; *López-Sánchez and Frade, 2013*; *Frade and Ovejero-Benito, 2015*) and cell cycle exit of a variety of tumor cells (*Jin et al., 2007*) and glial cells (*Cragnolini et al., 2009*; *2012*). However, a role for p75NTR in regulating cell cycle exit has not been previously demonstrated in neuronal progenitors in the developing brain. We show that the absence of p75NTR led to a delay in cell cycle exit of GCPs, indicating that p75NTR is necessary for timely withdrawal from the cell cycle. The lack of p75NTR was sufficient to increase the size of the cerebellum, a difference that persisted into adulthood, compromising the normal motor/balance function of these animals. In addition, we investigated the p75NTR ligands expressed in the cerebellum that elicited cell cycle arrest of GCPs, and demonstrate a specific role for proNT3 in blocking sonic hedgehog-induced proliferation.

Our results suggest that proNT-3 stimulation of p75NTR regulates the timing of cessation of proliferation of GCPs during cerebellar development, and reveal a novel role for p75NTR in developing neuronal progenitors.

## Results

### Developmental expression of p75$^{NTR}$ in the cerebellum

During development p75$^{NTR}$ is highly expressed in the EGL throughout the cerebellum (*Figure 1a, b*). As Purkinje cells develop, they also express p75$^{NTR}$, such that at P21 when the EGL is nearly gone, p75$^{NTR}$ remains expressed in Purkinje cells (*Figure 1a,b*). A Western blot of cerebellum lysates from different postnatal ages shows peak expression of p75$^{NTR}$ at P7 (*Figure 1c*). Although the major function for p75$^{NTR}$ that has been defined in the CNS is promoting neuronal death, especially after injury, few apoptotic cells were detected in the cerebellum in vivo in either WT of p75$^{NTR}$-/- (*Ngfr-/-*) mice (*Figure 1—figure supplement 1*), consistent with previous studies (*Carter et al., 2003*).

During early postnatal development, when most of the cells in the EGL are actively proliferating, nearly all the GCPs are positive for p75$^{NTR}$, but as development proceeds and the cells start to differentiate and migrate, two sub-populations can be distinguished in the EGL (*Figures 1b*, *2*). The external EGL (eEGL) where cells express p75$^{NTR}$ and proliferation markers such as Ki67 (*Figure 2a*) and the internal EGL (iEGL) where p75$^{NTR}$ expression is downregulated, the cells stop proliferating and express doublecortin (DCX) as they start the migratory process toward the IGL (*Figure 2b*). This clear boundary between proliferating cells expressing p75$^{NTR}$ and migrating cells that lack p75$^{NTR}$ expression suggests that this receptor may be involved in regulating the transition of GCPs from a proliferating to a migrating population.

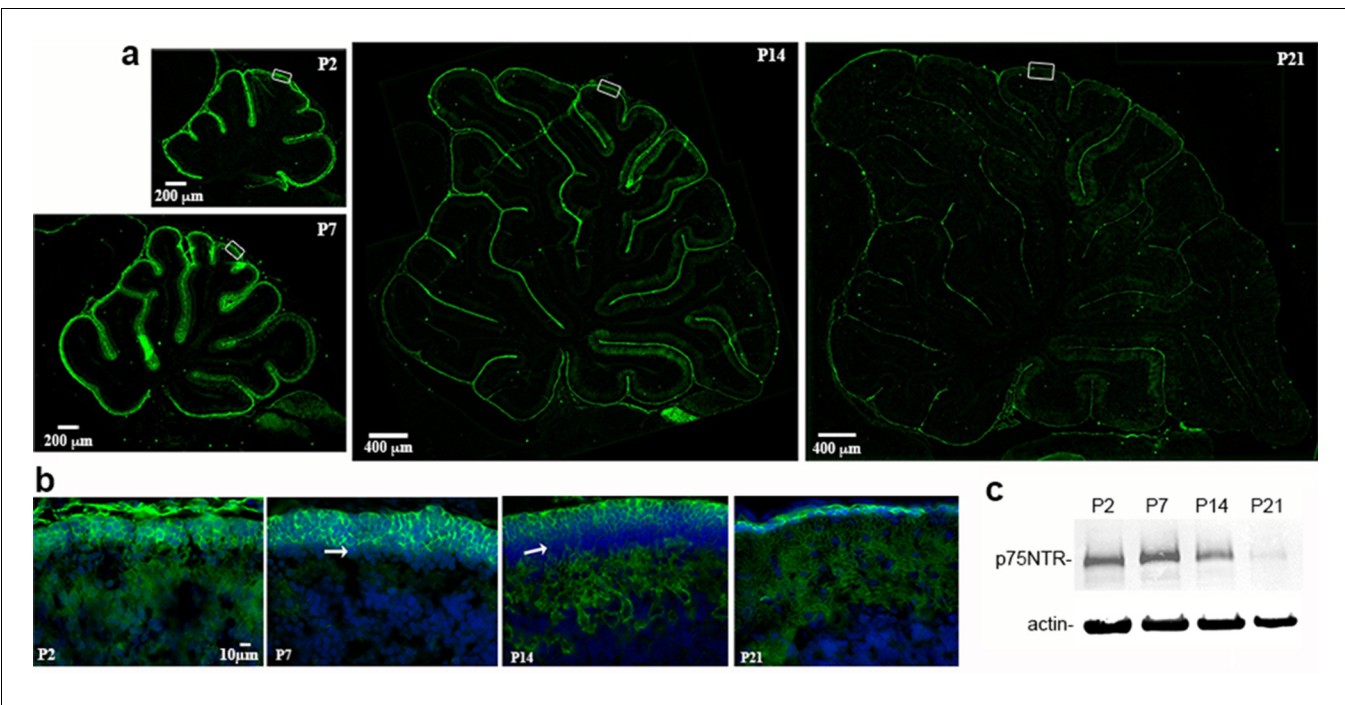

**Figure 1.** Development of p75$^{NTR}$ in the rat cerebellum. (a) Low magnification images of sagittal sections through the entire cerebellum from postnatal day (P) 2 through P21 showing the abundant immunolabeling for p75$^{NTR}$ in the EGL, which decreases by P21. Size bar indicates either 200 µm or 400 µm, as indicated. (b) High magnification images of lobe 6 showing p75$^{NTR}$ labeling in the outer EGL. Arrows indicate the inner EGL where the neurons lack p75$^{NTR}$. Size bar indicates 10 µm and is the same for all the images in B. (c) Western blot of cerebellum lysates from the indicated ages. Tissue was lysed with RIPA buffer containing protease inhibitors and 20 µg of total protein was separated on a 10% gel and probed for p75$^{NTR}$. The gel was re-probed for actin as a loading control and is representative of 3 independent experiments.

The following figure supplement is available for figure 1:

**Figure supplement 1.** No differences in TUNEL labeling in the EGL between *Ngfr-/-* and WT mice.

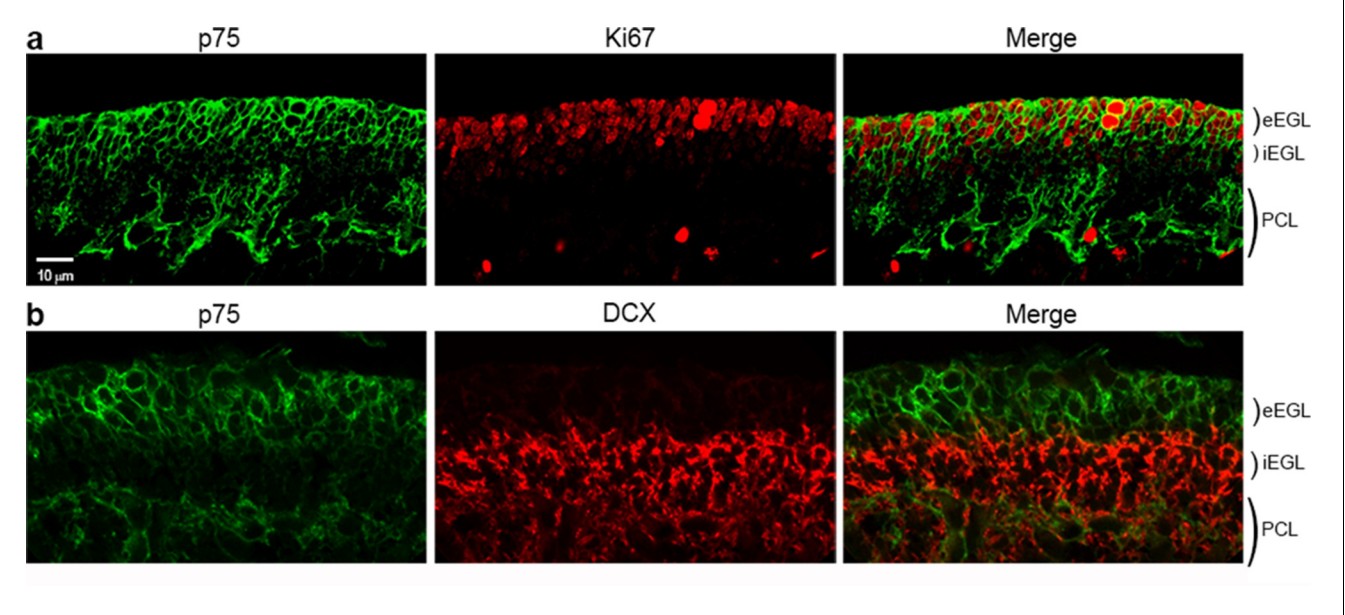

**Figure 2.** Expression of p75[NTR], Ki67, and DCX in the P7 rat cerebellum. (a) Confocal image of p75[NTR] and the proliferation marker Ki67 showing colocalization in the external EGL. Note that p75[NTR] is also expressed in developing Purkinje cells. (b) p75[NTR] is downregulated in the inner EGL when DCX is expressed. eEGL – external External Granule Layer, iEGL – inner External Granule Layer, PCL – Purkinje Cell Layer. Size bar is 10 µm.

BDNF is well known to promote survival and migration of granule cells via the TrkB receptor (*Borghesani et al., 2002*; *Segal et al., 1992*; *Courtney et al., 1997*). Consistent with its role in mediating survival and migration of GCPs, TrkB is highly expressed throughout the EGL, not just in the proliferative external zone where p75[NTR] is expressed, and is also present throughout the IGL (*Klein et al., 1990*; *Minichiello and Klein, 1996*). The differential expression of p75[NTR] and TrkB suggest that these receptors mediate distinct functions in the developing cerebellum.

### GCPs in p75[NTR]-/- (*Ngfr-/-*) mice have delayed cell cycle exit

The abundant expression of p75[NTR] in the EGL, colocalizing with proliferation markers, suggested a possible role in cell cycle regulation of GCPs. To assess this, EdU was injected at P5, P7, P10, and P14. In wild type mice the age of maximal GCP proliferation in the EGL is at P5, compared to P7 in the rat. Low magnification images show the overview of the cerebellum with similar EdU incorporation in WT and *Ngfr-/-* mice at P5, and the progressive decrease in EGL labeling in the WT beginning at P7, starting with the anterior lobes. EdU labeling remained high in the *Ngfr-/-* mice at P7 and P10 and only began to decrease at P14 (*Figure 3a*). GCPs in lobe 6 were the last to leave the cell cycle, and incorporation of EdU into cells in the EGL of this region was quantified. EdU labeling remained elevated in the *Ngfr-/-* mice compared with WT, with the difference most apparent at P10 and P14 (*Figure 3b,c*). To confirm these results, we compared expression of Ki67 in the EGL between wild type and *Ngfr-/-* mice at different developmental ages. At early postnatal ages (P2-P5), during maximal proliferation, there was no difference in the number of cells that expressed Ki67 between WT and *Ngfr-/-* mice (*Figure 3d,e*). However, at P7 in the EGL of WT mice, as cells began to withdraw from the cell cycle, there was a consequent reduction in the number of Ki67-positive cells (*Figure 3e*). In contrast, this reduction was not observed in the *Ngfr-/-* mice, where the number of proliferating cells remained high. As with EdU incorporation, this difference persisted and was even more evident at P10 and P14, when few proliferating GCPs remained in the EGL of WT mice. Cerebellar tissue taken from WT and *Ngfr-/-* mice at different developmental ages also showed a clear difference in the level of cyclin E1 at P10 and P14 (*Figure 3f*). These data indicated that the absence of p75[NTR] resulted in continued GCP proliferation in the EGL, and support a potential role for p75[NTR] in regulating the proper timing of cell cycle exit of GCPs.

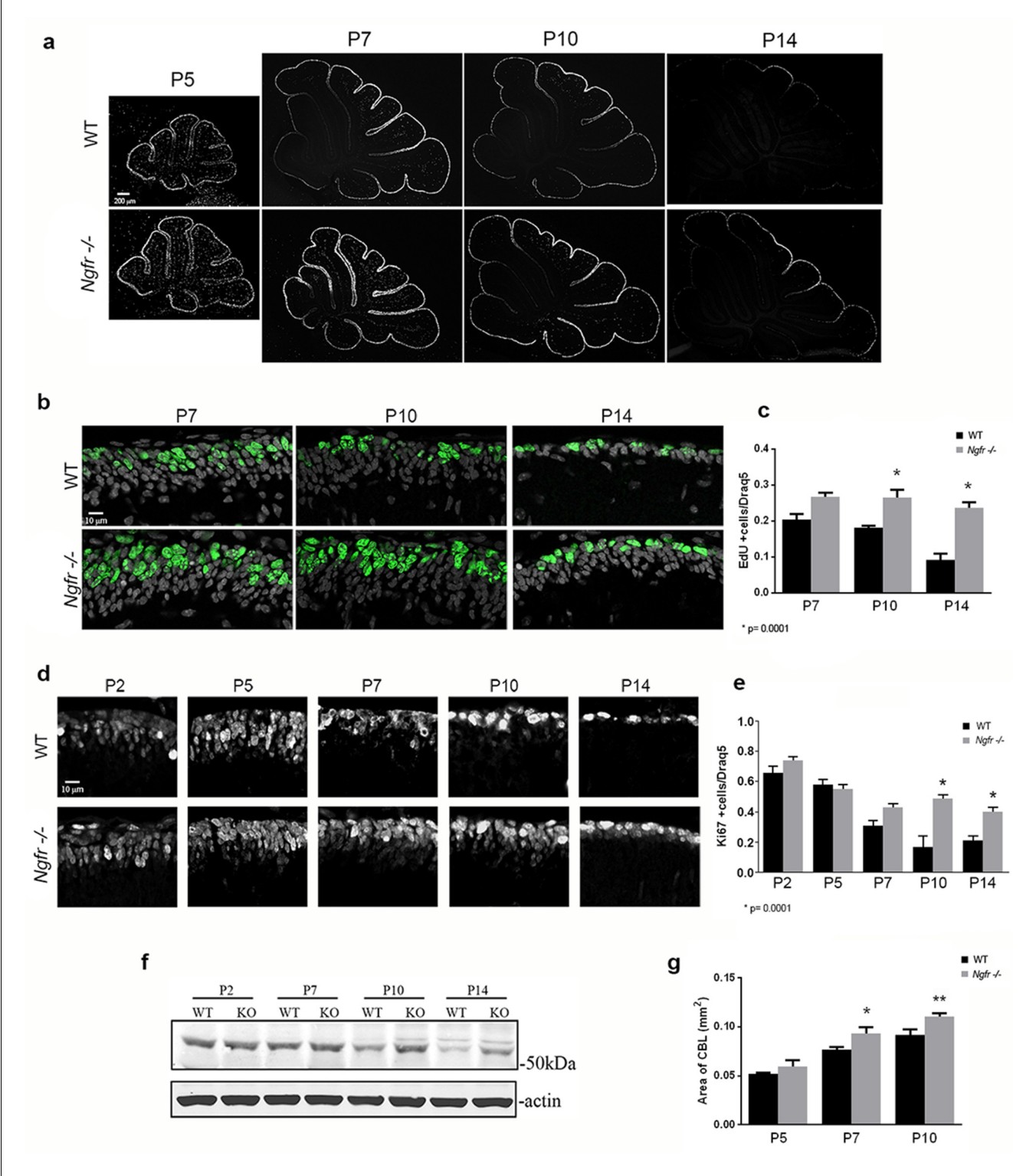

**Figure 3.** Cell cycle withdrawal of GCPs in the EGL is delayed in the *Ngfr-/-* mice compared to wild type mice. (a) Low magnification images of the entire cerebellum showing incorporation of EdU at postnatal ages from P5 to P14. GCPs in WT mice began to decrease EdU incorporation at P7, which continued decreasing at P10 and P14 as the progenitors left the cell cycle. Mice lacking p75[NTR] continued to incorporate EdU at high levels at P7 and P10, and only began to decrease proliferation at P14. Size bar is 200 μm. (b) High magnification of EdU labeling in lobe 6 showing continued EdU incorporation *Ngfr-/-* mice compared to WT from P7 through P14. Nuclei labeled with Draq5 are shown in gray. Size bar indicates 10 μm. (c) Quantification of EdU labeling of WT and *Ngfr-/-* mice. EdU-labeled cells were counted across 150 μm in the EGL of lobe 6b and are graphed relative

*Figure 3 continued on next page*

*Figure 3 continued*

to the total number of cells labeled with Draq5. At least three brains per genotype were analyzed at each age. *significantly different from WT at p=0.0001. (**d**) Developmental expression of Ki67 from P2 through P14, confirming the increased expression of proliferation markers in the *Ngfr-/-* mice compared to WT from P7 thorough P14. Size bar indicates 10 μm. (**e**) Quantification of cells expressing Ki67 in lobe 6b. Labeled cells were counted across 150 μm in the EGL of lobe 6b and are graphed relative to the total number of cells labeled with Draq5. At least three brains per genotype were analyzed at each age. *significantly different from WT at p=0.0001. Data in graphs in **c** and **e** are expressed as mean +/- SEM of at least three independent experiments. Asterisks indicate significance by ANOVA with Tukey's posthoc analysis, p values indicated below each graph. (**f**) Western blot showing the comparison of cyclin E1 expression in cerebellar lysates from WT or *Ngfr-/-* mice at the indicated postnatal ages. Differences in cyclin E levels between WT and knockout mice are evident at P7, P10 and P14. Blot is representative of three independent experiments. (**g**) Progressive increase in area of the cerebellum in *Ngfr-/-* mice compared to WT at P5, P7 and P10. *P7 *Ngfr-/-* significantly different from P7 WT at 0.038 by t-test **P10 *Ngfr-/-*significantly different from P10 WT at p=0.0174 by t-test.

The following source data is available for figure 3:

**Source data 1.** Mean number of labeled cells of 3 slides for each animal, and statistical analysis for the graphs shown in 3c, 3e, and 3g.

To determine whether the continued proliferation of GCPs in the absence of p75$^{NTR}$ resulted in a larger cerebellum, we compared the area of the cerebellum of WT and *Ngfr-/-* mice in mid-sagittal sections. At P5 no difference in size was observed, but at P7 and P10 the size of the cerebellum of *Ngfr-/-* animals was larger compared to WT animals (*Figure 3g*). This progressive difference in size corresponds to the timing of the differences we observed in expression of proliferation markers, becoming apparent when the GCPs of WT mice began to withdraw from the cell cycle at P7, while the GCPs of the *Ngfr-/-* mice continued to proliferate.

## ProNT-3 promotes cell cycle arrest of GCPs

p75$^{NTR}$ can respond to many different ligands by associating with distinct co-receptors. When it forms a complex with Trk receptors, p75$^{NTR}$ can facilitate responses to mature neurotrophins, however in association with a member of the sortilin family p75$^{NTR}$ binds proneurotrophins. To identify which ligand(s) might act via p75$^{NTR}$ to promote cell cycle exit of GCPs, granule cells from P7 rat cerebella were cultured and exposed to Shh with or without the different mature or proneurotrophins in the presence of BrdU to evaluate effects on proliferation.

Shh is a well-established mitogen for GCPs, and induced an increase in BrdU incorporation in GCP cultures, as expected. Incubation of cultured GCPs with any of the mature neurotrophins, NGF, BDNF, NT3, or NT4 had no effect on either the basal level of proliferation or Shh-induced BrdU incorporation, even at a dose of 100 ng/ml at which they can bind p75$^{NTR}$ (*Dechant et al., 1994*) (*Figure 4*). However, when the proneurotrophins were tested, proNT3 completely prevented the Shh-induced increase in BrdU incorporation (*Figure 5a,b*). This inhibition was reversed by anti-proNT3 (*Figure 5c*). Since activation of p75$^{NTR}$ by proneurotrophins is known to promote apoptosis in neurons (*Lee et al., 2001*; *Volosin et al., 2008*), TUNEL assay and cleaved caspase 3 staining were performed on cultured GCPs treated with proNT3. No significant difference was observed when cultured neurons were incubated with or without proNT-3 (*Figure 5 —figure supplement 1*), suggesting that the difference in BrdU incorporation was due to cell cycle arrest and not apoptosis of GCPs. Neither proNGF nor proBDNF had any effect on BrdU incorporation induced by Shh (*Figure 5a*), suggesting that this was a specific effect of proNT-3. Both proNGF and proBDNF, as well as proNT3, induced death of cultured hippocampal neurons at the doses tested, confirming that they were functional (*Figure 5—figure supplement 2*).

To confirm that the inhibition of Shh-induced BrdU incorporation by proNT3 was mediated through p75$^{NTR}$, GCPs were cultured from WT or *Ngfr-/-* mice. In the cells lacking p75$^{NTR}$, proNT-3 was unable to reduce the level of BrdU incorporation induced by Shh, demonstrating that proNT-3 required p75$^{NTR}$ to promote cell cycle exit of GCPs (*Figure 6a*). In addition, we assessed which member of the sortilin family could function as a co-receptor for proNT3 in regulating GCP proliferation. Only SorCS2 was detected in the EGL (*Figure 6b*). To assess whether SorCS2 was the functional co-receptor for proNT3, cultured GCPs treated with Shh+proNT3 were exposed to anti-SorCS2, which blocked the ability of proNT3 to decrease BrdU incorporation (*Figure 6c*). These data indicate that SorCS2, as well as p75$^{NTR}$, was required for the anti-proliferative actions of proNT3.

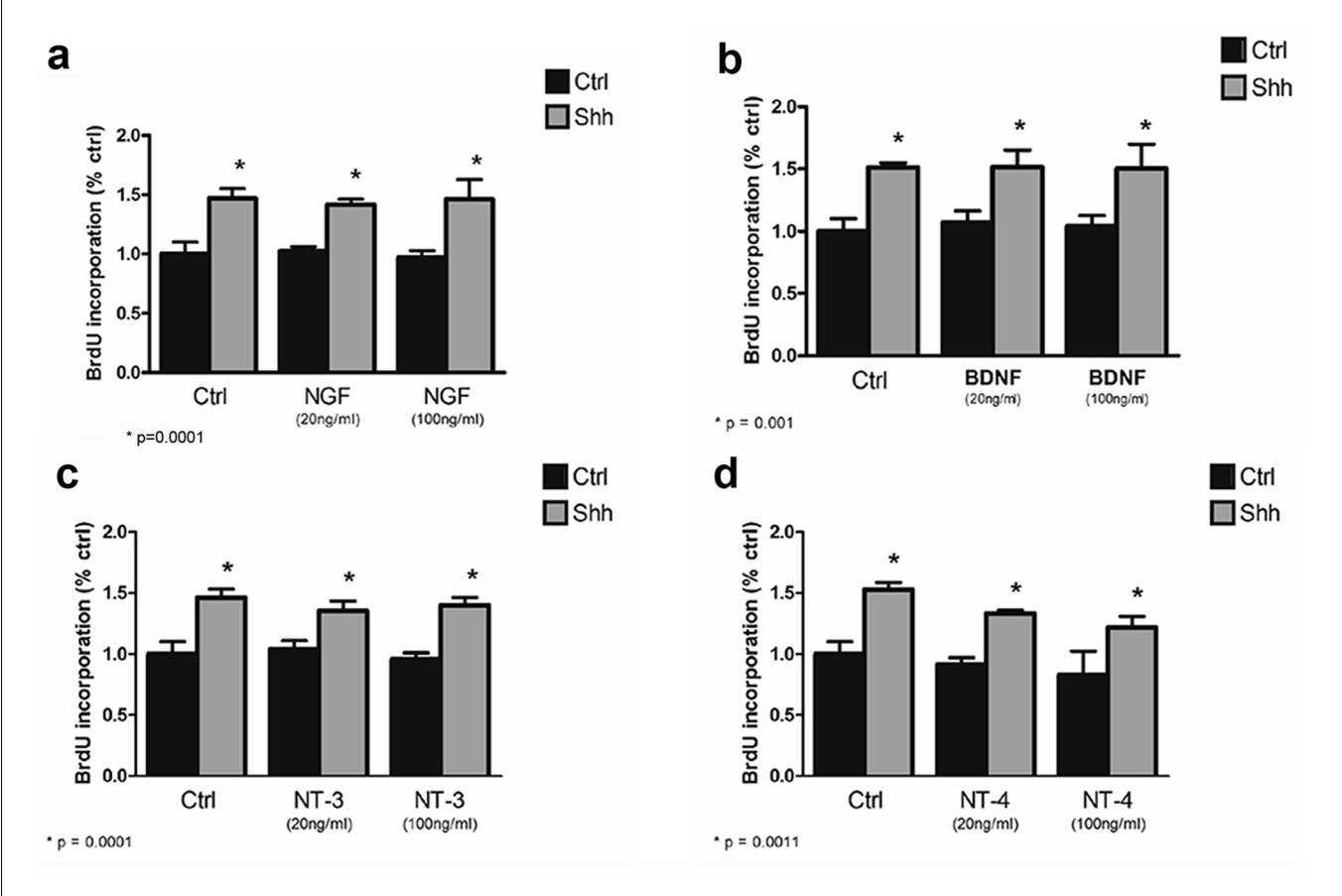

**Figure 4.** Mature neurotrophins have no effect on proliferation of cultured P7 rat GCPs in the absence or presence of Shh. GCPs from P7 rat were cultured with BrdU for 48 hr in the absence or presence of Shh with or without the different mature neurotrophins. BrdU labeling was analyzed by in-cell Western on the LiCor Odyssey. (a) GCPs with or without 20 ng/ml or 100 ng/ml of NGF. (b) GCPs with or without 20 ng/ml or 100 ng/ml of BDNF. (c) GCPs with or without 20 ng/ml or 100 ng/ml of NT3. (d) GCPs with or without 20 ng/ml or 100 ng/ml of NT4. Data are expressed as mean +/- SEM from three independent experiments. Asterisk indicates that Shh-treated cells increased BrdU incorporation compared to controls by ANOVA with Tukey's posthoc analysis, p values are indicated below each graph.

The following source data is available for figure 4:

**Source data 1.** Mean values for each experiment and statistical analysis for all graphs.

ProNT3 was the only factor of all the mature and proneurotrophins that prevented Shh-induced proliferation of GCPs, therefore we investigated whether proNT3 was expressed in the developing cerebellum. Immunostaining with an antibody to the pro domain revealed expression of proNT3 in Purkinje cells (*Figure 7a*), appropriately localized to affect GCPs in the internal EGL where they stop proliferating prior to migrating to the IGL.

Since proNT3 is the precursor to mature NT3, which is highly expressed in Purkinje cells (*Zhou and Rush, 1994*; *Friedman et al., 1998*), it was critical to ascertain whether uncleaved proNT3 could be secreted in the cerebellum. Cultures were prepared from P7 cerebellum and media was collected from cells depolarized with 25 mM KCl or undepolarized, and analyzed by immuno-precipitation followed by Western blot. ProNT3 was detected in the media with or with KCl treatment, indicating that cerebellar neurons can secrete proNT3, and the release was not dependent on activity (*Figure 7b*). To validate the specificity of the anti-proNT3 antibody, immunoprecipitation and immunostaining were performed on brain tissue from newborn *Ntf3-/-* mice and compared to tissue from *Ntf3+/+* and *Ntf3+/-* mice (*Figure 7 —figure supplement 1*).

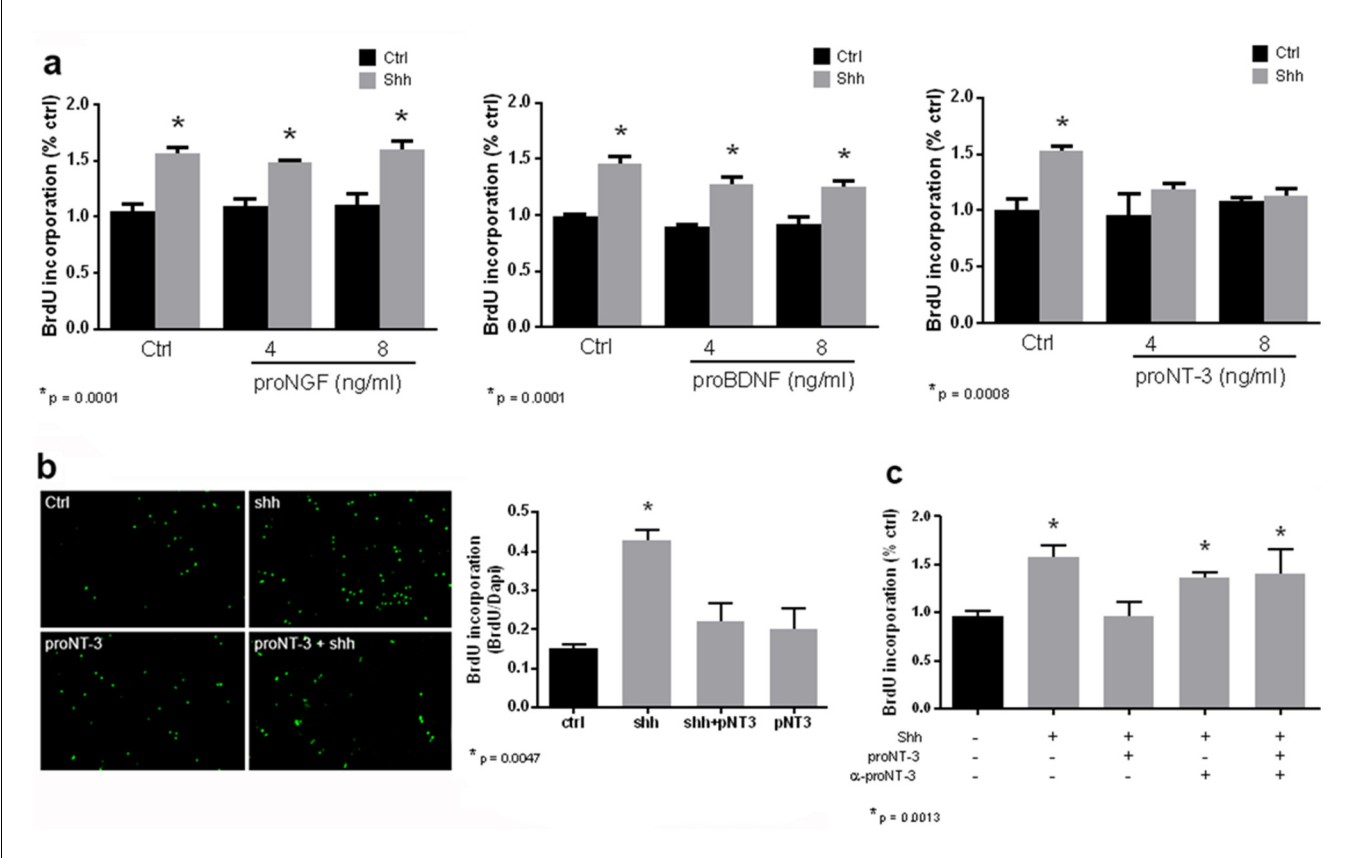

**Figure 5.** ProNT3, but not proNGF or proBDNF, prevented Shh-induced proliferation of cultured GCPs. (**a**) GCPs were cultured from P7 rat cerebella with BrdU in the absence or presence of Shh with or without 4 or 8 ng/ml of proNGF, proBDNF or proNT3 for 48 hr. Cells were analyzed by in-cell Western for BrdU incorporation. (**b**) P7 rat GCPs were cultured for 48 hr with BrdU in the absence or presence of Shh, proNT3, or Shh + proNT3. Cells were fixed and immunostained for BrdU, and the number of labeled cells was counted, shown in the graph. (**c**) P7 rat GCPs were cultured without or with Shh, Shh+proNT3, or Shh+proNT3+anti-proNT3 for 48 hr and analyzed by in-cell Western for BrdU incorporation. Data in the graphs are expressed as mean values +/- SEM from at least 3 independent experiments. Asterisks indicate significance by ANOVA with Tukey's posthoc analysis, with the indicated p value below each graph.

The following source data and figure supplements are available for figure 5:

**Source data 1.** Mean values for each experiment and statistical analysis for all graphs in *Figure 5*.

**Figure supplement 1.** ProNT3 does not induce apoptosis of cerebellar neurons.

**Figure supplement 2.** Proneurotrophins induced death of hippocampal neurons.

## Stimulation of p75[NTR] decreases Shh induction of HDAC1

Shh signaling requires HDAC1 activity to maintain GCPs in a proliferating and undifferentiated state (*Canettieri et al., 2010*). As GCP proliferation decreased starting at P7 in WT mice, the level of HDAC1 was reduced in the EGL. However, the GCPs in the *Ngfr-/-* mice continued proliferating, maintaining high levels of HDAC1, even at P14 (*Figure 8a*). Not only were there more GCPs expressing HDAC1, consistent with more proliferating cells in the EGL of *Ngfr-/-* mice, the intensity of HDAC1 expression was also higher in the knockout mice (*Figure 8b,c*). To determine whether stimulation of p75[NTR] could regulate levels of HDAC1, cultured cerebellar granule cells were treated with Shh with or without proNT3. Shh induced HDAC1, as expected, however treatment with proNT3 eliminated the Shh-mediated induction of HDAC1 within one hour of treatment (*Figure 8d,e*), indicating that activation of p75[NTR] was able to block a critical component of Shh signaling. HDAC1-

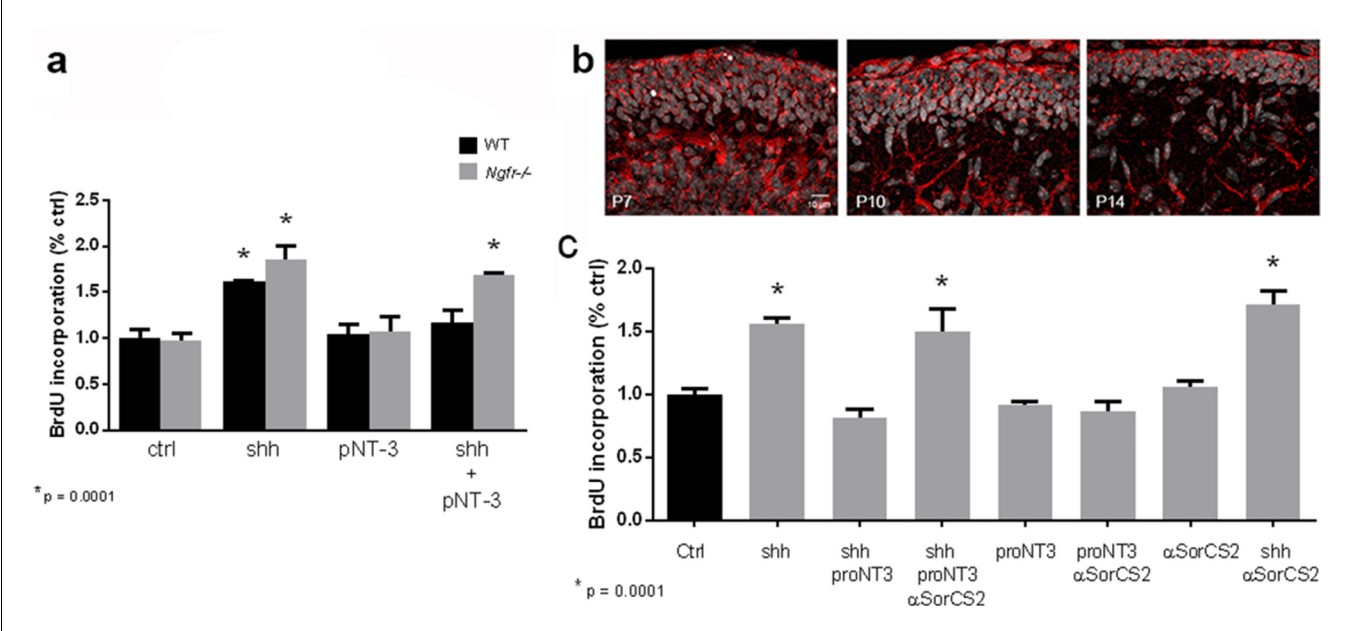

**Figure 6.** ProNT3 requires p75[NTR] and SorCS2 to block Shh-induced GCP proliferation. (**a**) GCPs from wild type or *Ngfr-/-* mice were cultured with Shh, proNT3 or Shh+proNT3 for 48 hr in the presence of BrdU and analyzed by in-cell Western analysis of anti-BrdU. (**b**) Immunostaining for SorCS2 demonstrates expression of this co-receptor in the EGL, shown for P7, P10 and P14. Size bar indicates 10 μm. (**c**) GCPs from P7 rat were cultured with Shh, proNT3 and anti-SorCS2. Anti-SorCS2 reversed the effects of proNT3 on Shh-induced BrdU incorporation, but had no effect by itself or with either proNT3 or Shh alone. Data in the graphs are expressed as mean values +/- SEM from at least 3 independent experiments. Asterisks indicate significantly different from control by ANOVA with Tukey's posthoc analysis, with p=0.0001.

The following source data is available for figure 6:

**Source data 1.** Mean values for each experiment and statistical analysis for graphs in 6A and 6c.

mediated deacetylation of Gli2 is necessary for this transcription factor to translocate to the nucleus and induce Gli1 mRNA (*Canettieri et al., 2010*), which leads to induction of additional genes needed to promote proliferation. Since proNT3 decreased levels of HDAC1, we investigated whether proNT3 also prevented induction of Gli1 mRNA by Shh. Cultured cerebellar neurons treated with Shh with or without proNT3 for 24 hr were analyzed by qPCR for Gli1 mRNA, and showed that proNT3 reduced Gli1 mRNA induction by Shh (*Figure 8f*).

## Effects of EGL-specific p75[NTR] (*Ngfr*) deletion

p75[NTR] is expressed in many brain regions during development, and even in the cerebellum this receptor is expressed in Purkinje cells as well as the GCPs in the EGL (*Figure 1*). To determine the specific role of p75[NTR] in the EGL, we generated mice that lack p75[NTR] expression in the EGL by mating floxed *Ngfr* mice (*Ngfr* [fl/fl]) (*Bogenmann et al., 2011*) with *Atoh1(Math1)*-CRE mice (Jackson labs). These mice retained p75[NTR] expression in Purkinje cells and the meninges but lacked p75[NTR] in the EGL (*Figure 9a*). Analysis of Ki67 expression in these animals demonstrated that the absence of p75[NTR] specifically from the GCP population was sufficient to increase the number of proliferating GCPs at P7, P10, and P14, indicating a delay in cell cycle withdrawal in the EGL, similar to what we observed in the global *Ngfr-/-* mice (*Figure 9b*). In addition to Ki67 expression, we also analyzed EdU incorporation and similarly observed higher level of incorporation at P7, P10, and P14 in the *Ngfr*[fl/fl]:*Atoh1*-Cre mice compared to floxed mice without Cre recombinase, which were identical to WT (*Figure 9c*).

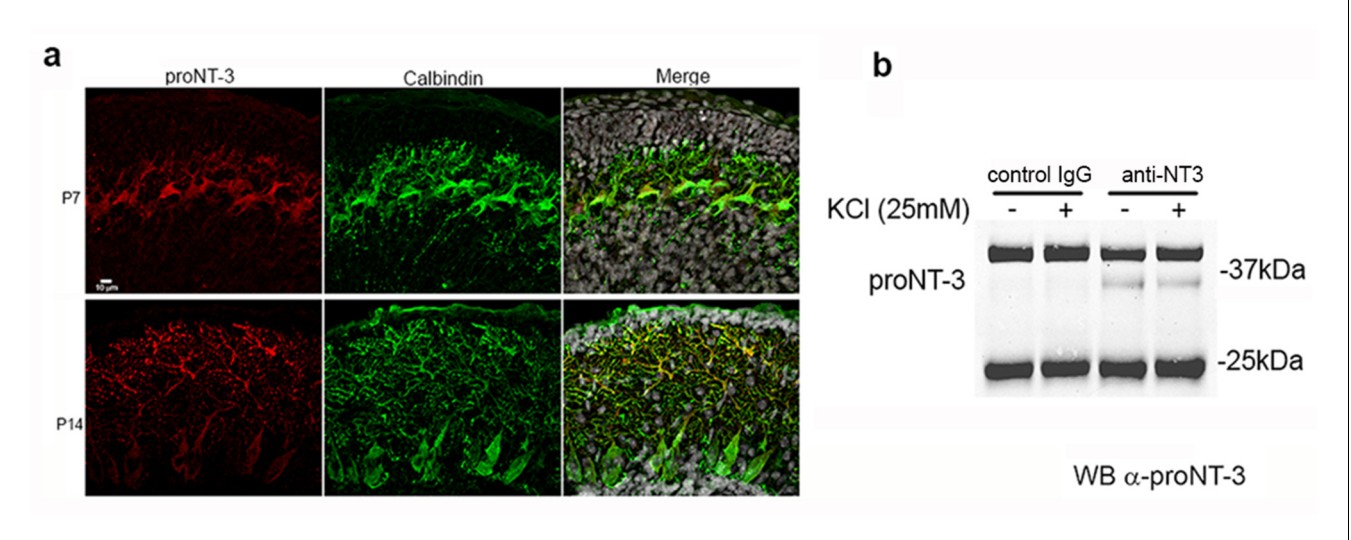

**Figure 7.** Expression and secretion of proNT3 in cerebellum. (a) Immunostaining of P7 and P14 rat cerebellum with an antibody to the pro domain of proNT3 shows the presence of proNT3 in Purkinje cells, labeled with calbindin. Note the abundant proNT3 labeling in the dendrites, especially apparent at P14. Nuclei labeled with Draq5 are shown in gray in the merged image. (b) Cultures of P7 rat cerebellum were treated with or without 25 mM KCl to depolarize the cells, and the media was analyzed by immunoprecipitation for NT3 followed by Western blot for proNT3, demonstrating that proNT3 can be secreted from cerebellar cells.

The following figure supplement is available for figure 7:

**Figure supplement 1.** Validation of the proNT3 antibody.

## Persistent effects into adulthood

Since we observed continued proliferation of GCPs during the development of the cerebellum when p75[NTR] was removed either globally or specifically from the EGL, with an increase in cerebellar size (*Figure 3g*), we sought to determine whether there was any persistent consequence for cerebellar size and function in adult animals. The increased cerebellar size in both the global and EGL-specific (*Ngfr*[fl/fl]:*Atoh1*-Cre) p75[NTR-/-]mice persisted into adulthood (*Figure 10a*). To further elucidate a possible functional consequence of these developmental changes, we analyzed the motor/balance coordination in these animals using the rotarod test. Animals were trained 5 times at slow speed (10 rpm) for 60 s with a 60 s recovery break between each trial. Animals were tested the next day with the same time interval for test/recovery but with increasing speeds of 5 rpm per trial until 40 rpm was reached. As shown in *Figure 10b*, the global *Ngfr-/-* mice performed poorly even at low speed. The latency to fall was significantly faster compared to the WT mice. However, the *Ngfr*[fl/fl]:*Atoh1*-Cre mice that specifically lacked p75[NTR] in the developing EGL also showed significant motor deficits compared to WT mice beginning at 20 rpm, indicating that the absence of this receptor specifically from the EGL during development caused a persistent deficit in motor function in the adult.

## Discussion

Neurotrophins regulate multiple aspects of neuronal function that critically impact brain development. Of the neurotrophins, BDNF, NT3 and NT4 are highly expressed in the developing cerebellum (*Friedman et al., 1998*). BDNF has been demonstrated to act via TrkB to maintain survival of GCPs and promote their migration to the IGL (*Borghesani et al., 2002*; *Minichiello and Klein, 1996*; *Zhou et al., 2007*). Consistent with this function, TrkB is expressed throughout the EGL and IGL. NT3 also promotes survival of granule neurons and dendritic development of Purkinje cells via TrkC (*Bates et al., 1999*; *Joo et al., 2014*). However, whether proneurotrophins play a role in cerebellar development has not been extensively investigated. p75[NTR] is abundantly expressed in granule cells only in the EGL, specifically in the outer proliferative layer, which gets progressively smaller with

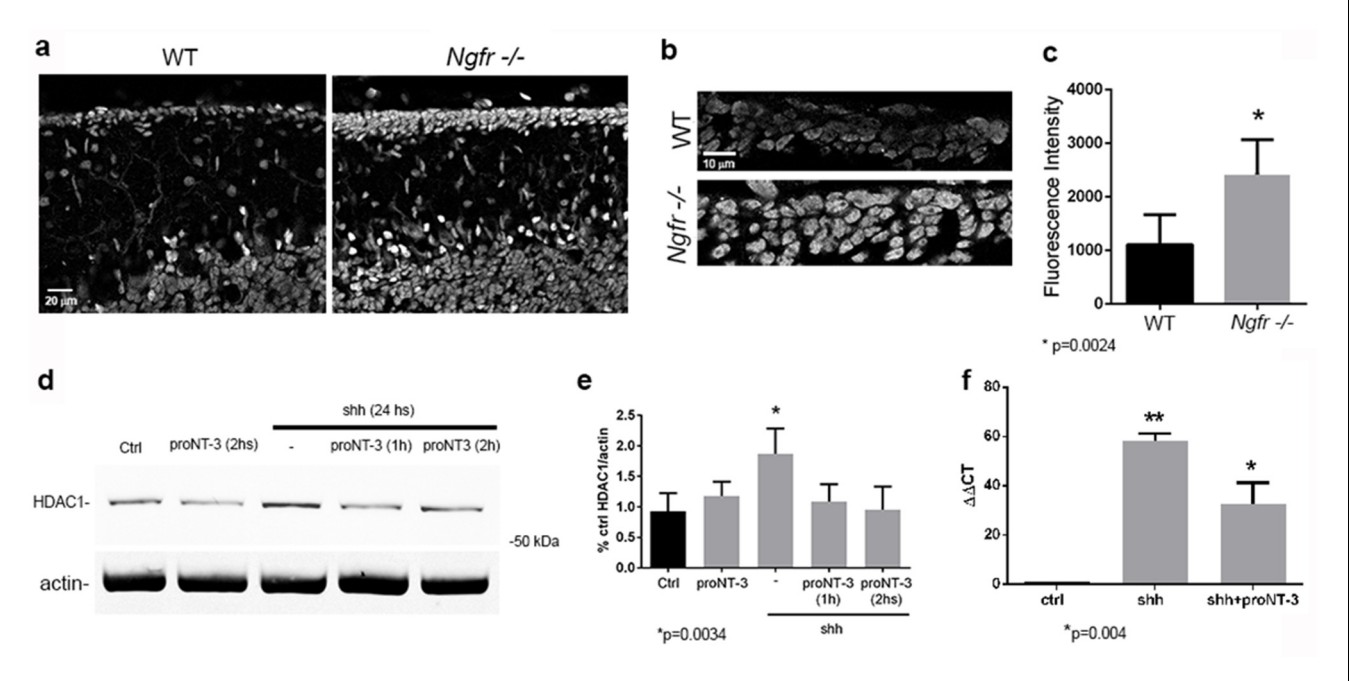

**Figure 8.** HDAC1 expression and regulation by Shh and proNT3. (**a**) Immunostaining for HDAC1 in WT and *Ngfr-/-* mice at P14. Size bar is 20 µm. (**b**) High magnification images of HDAC1 staining in the EGL in WT and *Ngfr-/-* mice at P14. Size bar is 10 µm. (**c**) Quantification of fluorescence intensity (mean gray value) of HDAC1 staining lobe 6b of the EGL at P14. *indicates significance at p=0.0024 by student's t-test, 6 brains of each genotype were analyzed. (**d**) Western blot of cultured GCPs from P7 rat cerebellum treated as indicated and probed for HDAC1. (**e**) Quantification of Western blots from 3 independent experiments showing that treatment with Shh increased HDAC1 expression, which was reduced by proNT3 within 1 hr. ProNT3 alone had no effect on HDAC1 expression. * indicates significantly different from control at p=0.0034. (**f**) Regulation of Gli1 mRNA by Shh and Shh +proNT3. ** indicates significantly different from control, * indicates significantly different from Shh alone, p=0.004 by ANOVA with Tukey's posthoc analysis.

The following source data is available for figure 8:

**Source data 1.** Mean values for each experiment and statistical analysis for all graphs in *Figure 8*.

development as the progenitors cease proliferating and migrate to the IGL. The p75[NTR] can bind many different ligands, associate with different co-receptors, and recruit a panoply of intracellular binding proteins to activate specific signaling pathways (*Charalampopoulos et al., 2012*; *Roux and Barker, 2002*), and therefore can influence multiple functions depending on the cellular context. The most well characterized effect mediated by p75[NTR] is cell death (*Frade et al., 1996*; *Friedman, 2000*), especially after injury such as that induced by seizures (*Roux et al., 1999*; *Troy et al., 2002*; *Unsain et al., 2008*). However, during development, p75[NTR] is widely expressed in the brain (*Yan and Johnson, 1988*), and is particularly abundant in the EGL of the cerebellum, where it does not mediate apoptosis. In the present work we demonstrate a novel function for p75[NTR], showing that this receptor is necessary for proper cell cycle exit of GCPs in the cerebellum, and that proNT3 can antagonize Shh-induced proliferation of GCPs and promote cell cycle withdrawal.

## Regulation of GCP proliferation

Sonic hedgehog (Shh) is a highly efficacious mitogen for GCPs and is vital for their clonal expansion within the EGL (*Dahmane et al., 1999*; *Wechsler-Reya and Scott, 1999*). Shh continues to be expressed during cerebellar maturation, which raises the question of how some GCPs in the EGL stop responding to Shh and begin to migrate to the IGL while others keep proliferating. Previous studies have shown that P5-6 is the time of maximal GCP proliferation in mice (*Roussel and Hatten, 2011*), and our results show that this coincides with the peak of p75[NTR] expression in the EGL. At this stage, almost all cells in the EGL were positive for both p75[NTR] and proliferation markers such as

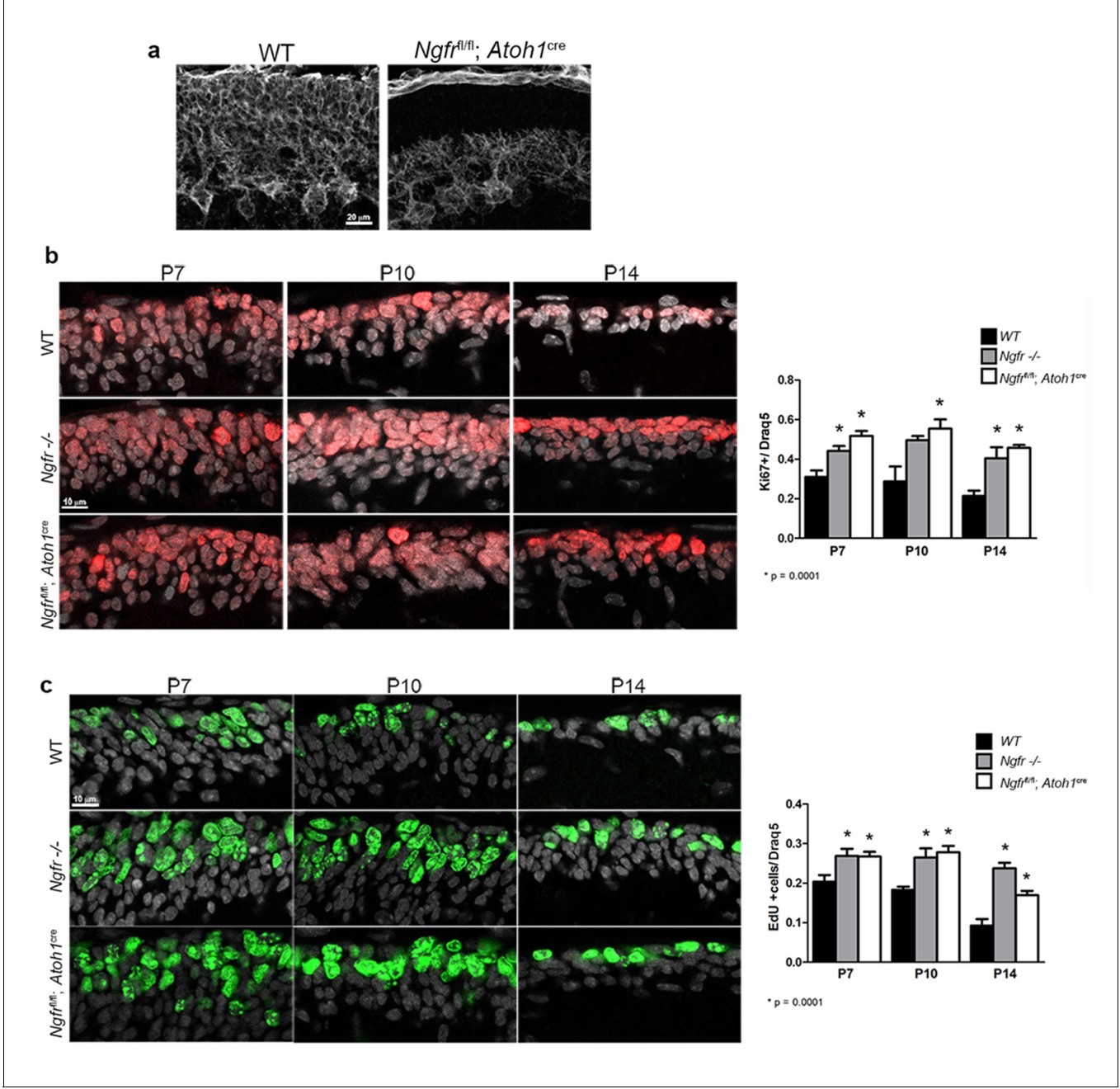

**Figure 9.** Specific deletion of p75[NTR] from the EGL elicits increased GCP proliferation. (**a**) *Ngfr*[fl/fl] mice crossed with the *Atoh1*-Cre show lack of p75[NTR] specifically in the EGL while retaining p75[NTR] expression in Purkinje cells and meninges. Size bar indicates 20 μm. (**b**) Both *Ngfr-/-* and *Ngfr*[fl/fl]-*Atoh1*-Cre mice show increased Ki67 labeling in the EGL at P7, P10 and P14. Total nuclei labeled with Draq5 are shown in gray. Labeled cells were counted across 150 μm in the EGL of lobe 6b and are graphed relative to the total number of cells labeled with Draq5. (**c**) EdU labeling showing increased incorporation at P7, P10, and P14 in both *Ngfr-/-* and *Ngfr*[fl/fl]-*Atoh*1-Cre mice. Total nuclei labeled with Draq5 are shown in gray. Labeled cells were counted across 150 μm in the EGL of lobe 6b and are graphed relative to the total number of cells labeled with Draq5. At least three mice per genotype were analyzed at each age. Size bars in B and C indicate 10 μm. Data in all graphs are expressed as mean +/- SEM of at least three independent experiments. Asterisks indicate significantly different from WT by ANOVA with Tukey's posthoc analysis with the p values below each graph.

The following source data is available for figure 9:

**Source data 1.** Mean number of labeled cells of 3 slides for each animal, and statistical analysis for all graphs in *Figure 9*.

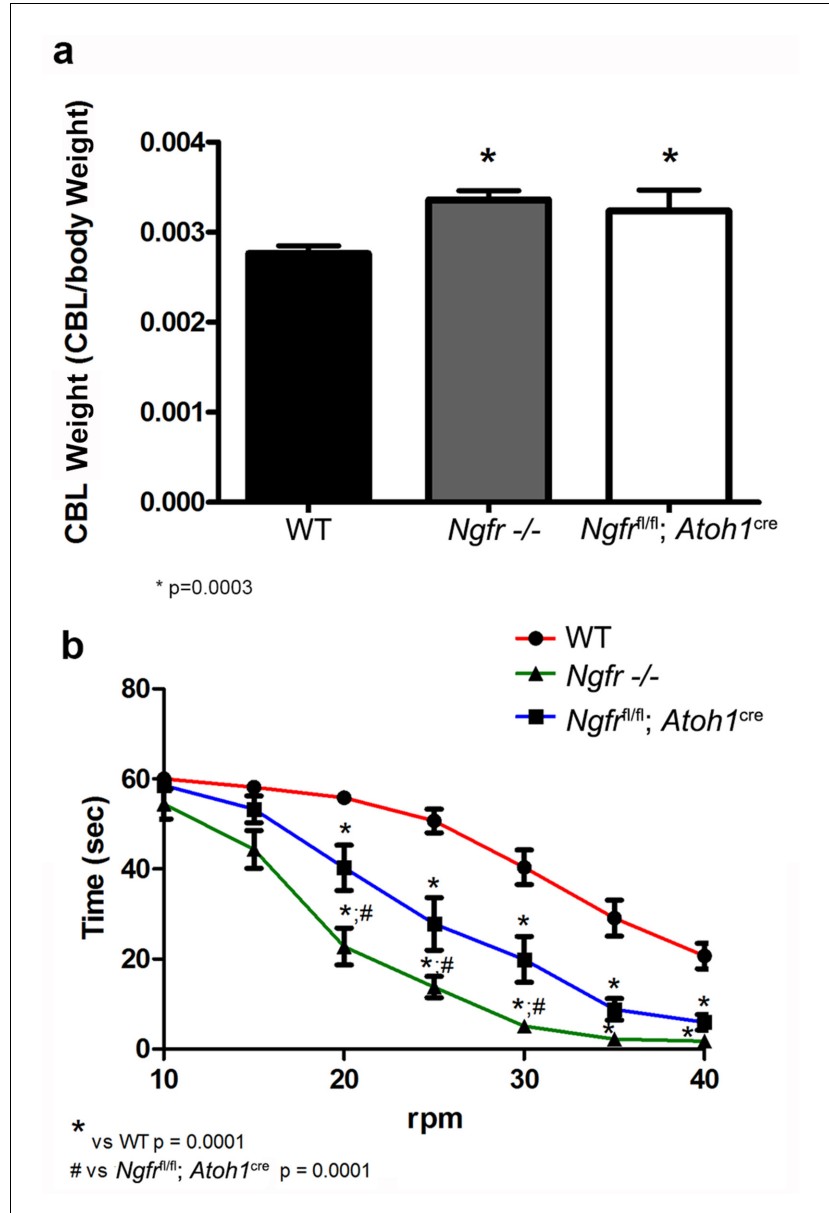

**Figure 10.** The absence of p75NTR during EGL development has persistent effects in the adult. (a) Cerebellar size of both *Ngfr-/-* and *Ngfr*fl/fl-*Atoh1*-Cre was increased in the adult compared to WT mice. * indicates significance at p=0.0003. (b) Motor balance on the rotarod was impaired in both *Ngfr-/-* and *Ngfr* fl/fl-*Atoh1*-Cre. At least eleven mice per genotype were tested. Asterisk indicates significantly different from WT, # indicates *Ngfr -/-* mice performed significantly worse than *Ngfr*fl/fl-*Atoh1*-Cre, p=0.0001 by ANOVA with Tukey's posthoc analysis.

The following source data is available for figure 10:

**Source data 1.** Mean values for each experiment and statistical analysis for all graphs in *Figure 10*.

Ki67 and incorporation of EdU. After P7, two sublayers became apparent in the EGL. In the external zone, GCPs continued to proliferate and express p75NTR, while cells in the internal layer of the EGL no longer expressed p75NTR or proliferation markers as they began to differentiate and migrate tangentially in the medial-lateral plane (*Legué et al., 2015*) and radially towards the IGL (*Wang and Zoghbi, 2001*). In WT mice the number of proliferating GCPs in the EGL began to decrease at P7, and continued to decrease until few proliferating cells remained in the EGL at P14. In contrast, in

both the global *Ngfr-/-* mice and the mice specifically lacking p75[NTR] in the EGL (*Ngfr*[fl/fl]:*Atoh1*-Cre mice), the number of proliferating, Ki67-positive cells remained elevated at P7, P10, and P14. However, the granule cells eventually ceased proliferating, and no tumors formed in these mice. These results suggest that there is a delay in cell cycle exit of GCPs lacking p75[NTR], sufficient to cause an enlarged cerebellum that persisted into adulthood, and defects in motor coordination in these animals. Since no tumors formed in these mice and the GCPs in the *Ngfr-/-* mice eventually stopped proliferating, it is likely that other anti-mitotic factors caused the cells to eventually leave the cell cycle. Several factors have been identified that promote cell cycle exit, including BMP2 and 4 (*Rios et al., 2004*), Wnt3 (*Anne et al., 2013*), and PACAP (*Nicot et al., 2002*), each acting by a distinct mechanism to block Shh activation of the Gli transcription factors. However, despite the presence of these other factors, the specific absence of p75[NTR] caused the GCPs to continue proliferating beyond the usual developmental period, indicating that this receptor is important for the timing of cell cycle withdrawal. A question that remains to be addressed is whether the different anti-proliferative factors have redundant effects to ensure normal development of the cerebellum, or if each factor is necessary for maturation of different subpopulations of granule cells, all of them necessary for proper function. We observed that the expression of p75[NTR] persisted the longest in lobes VIb, VII, and VIII, consistent with a previous study (*Carter et al., 2003*) and in the same regions where we detected proliferation persisting the longest in the EGL, however the delay in cell cycle exit of GCPs in both the global *Ngfr-/-* and the *Ngfr*[fl/fl]:Atoh-Cre mice was evident throughout the cerebellum.

The p75[NTR] has previously been shown to regulate cell cycle of Schwann cells by recruiting an adapter protein, SC1, which transcriptionally represses cyclin E (*Chittka et al., 2004*). We have previously shown that p75[NTR] can also repress cyclin E in proliferating astrocytes (*Cragnolini et al., 2012*), suggesting that cyclin E may be a target of p75[NTR] anti-proliferative signaling. Interestingly, we detected a clear difference in cyclin E1 expression in the cerebellum of *Ngfr-/-* mice compared to WT, which was particularly evident at P10 and P14, the ages showing the greatest differences in GCP proliferation.

## Regulation of HDAC1

Shh is expressed by Purkinje cells and regulates the proliferation of the GCPs (*Dahmane et al., 1999*). Shh signals by regulating the Gli transcription factors, specifically activating Gli2, which then targets Gli1 for transcriptional induction (*Blaess et al., 2006*; *Hui and Angers, 2011*). Gli1 and Gli2 are acetylated proteins, and their transcriptional activation requires deacetylation by HDAC1 (*Canettieri et al., 2010*), which is highly expressed in proliferating GCPs (*Yoo et al., 2013*). HDAC1 is also required for deacetylation of histones, a crucial developmental regulatory event leading to condensation of chromatin and transcriptional repression, maintaining the cells in an immature, undifferentiated state (*Dovey et al., 2010*). Previous studies have shown that HDAC activity is necessary for Shh-induced proliferation of cultured GCPs (*Lee et al., 2013*). Coincident with the continued proliferation of GCPs in the *Ngfr-/-* mice, we observed that HDAC1 expression remained elevated in the EGL compared to the WT mice, where HDAC1 decreased as the cells ceased proliferating. We demonstrated that Shh induced HDAC1 in cultured GCPs and that this induction was reversed by proNT3. ProNT3 also attenuated the induction of Gli1 mRNA by Shh, suggesting a mechanism by which p75[NTR] signaling may interfere with the Shh pathway by inhibiting HDAC1-mediated deacetylation of Gli2 to block Shh signaling, thereby promoting withdrawal from the cell cycle. Overexpression of Shh has previously been shown to result in an enlarged cerebellum due to overproduction of granule cells (*Corrales et al., 2004*). We have similarly observed an enlarged cerebellum in mice lacking p75[NTR] in the EGL, suggesting that removing this block of Shh signaling has a similar consequence as overexpression of Shh, resulting in the production of excess granule cells and an enlarged cerebellum.

## ProNT3 is an anti-proliferative ligand in the cerebellum

In this study we demonstrated for the first time that proNT3 is a negative regulator of GCP proliferation, acting via p75[NTR]. NT3 expression has previously been detected in the developing cerebellum, however the mRNA was localized by in situ hybridization to granule cells (*Lindholm et al., 1993*) and the protein has been localized to Purkinje cells (*Friedman et al., 1998*; *Zhou and Rush, 1994*).

In this study we demonstrate that proNT3 protein is also present in Purkinje cells, and can be secreted as the uncleaved proneurotrophin. Mature neurotrophins had no effect on GCP proliferation. However, since all the proneurotrophins can activate p75$^{NTR}$ we expected that any proneurotrophin would reduce Shh-induced proliferation of GCPs. Surprisingly, proNGF and proBDNF had no effect on GCP proliferation, although they effectively promoted death of hippocampal neurons. Thus, proNT-3 proved to be the only neurotrophin to promote cell cycle exit in this neuronal population, suggesting that there is specificity among the different p75$^{NTR}$ ligands for their ability to influence distinct cellular functions. The basis for this specificity is unknown, however p75$^{NTR}$ has been shown to bind many different ligands to trigger distinct responses. Even the same ligand binding to p75$^{NTR}$ on the same cells may trigger different responses depending on the state of the cell. For example, NGF treatment of cultured hippocampal neurons initially promotes neurite outgrowth via p75$^{NTR}$ (*Brann et al., 1999*) but later promotes p75$^{NTR}$-dependent neuronal death (*Friedman, 2000*). Additionally, how the pro domains of proneurotrophins influence p75$^{NTR}$ activity is not understood. Recent studies showed that the pro domain of proBDNF induced LTD in hippocampal slices via a p75$^{NTR}$-dependent mechanism, but the proNGF pro domain failed to do so (*Mizui et al., 2015*). Additionally, the BDNF pro domain induced growth cone retraction, which required p75$^{NTR}$ even though the pro domain does not directly bind p75$^{NTR}$, but rather binds to the co-receptor of the sortilin family, in this case SorCS2 (*Anastasia et al., 2013*). Interestingly, the pro domain of proBDNF has a naturally occurring Val66Met polymorphism. Although both the Val66 and Met66 pro domains bound the SorCS2 receptor, only the Val66 pro domain induced growth cone collapse, the Met66 pro domain did not (*Anastasia et al., 2013*), indicating that slight differences in the pro domain can lead to differences in functional outcome. Our current study also identified SorCS2 as the co-receptor for proNT3 actions on the GCP population, suggesting that the distinct pro domains of proneurotrophins differentially bind to the SorCS2 co-receptor and influence how p75$^{NTR}$ interacts with specific signaling pathways to regulate cellular responses, which may account for the specificity of proNT3 in regulating GCP cell cycle exit.

A previous study had suggested that p75$^{NTR}$ may mediate some of the effects of BDNF on cerebellar patterning and foliation. However, using the global *Ngfr*-/- mice this observation may have been largely due to effects in Purkinje cells, which also express p75$^{NTR}$, and knockout of p75$^{NTR}$ led to a worsening of Purkinje cell morphology seen with reduced BDNF levels (*Carter et al., 2003*). The expression of p75$^{NTR}$ in Purkinje cells is not likely to play a role in cell cycle regulation since these cells are already postmitotic, an additional indication that the function of p75$^{NTR}$ depends on the cell context. That study also showed that p75$^{NTR}$ did not appear to play a role in apoptosis in the developing cerebellum, consistent with our findings.

p75$^{NTR}$ is highly expressed throughout the brain during development. Even within the cerebellum this receptor is found in Purkinje cells as well as in GCPs. Moreover, p75$^{NTR}$ is developmentally expressed in many neuronal populations involved in motor function, such as spinal motor neurons (*Ernfors et al., 1989*), striatum, and cortex (*Yan and Johnson, 1988*). Not surprisingly, therefore, the global *Ngfr*-/- mice performed poorly on the rotarod test. To ascertain whether the changes in the granule cell population resulting from the specific lack of p75$^{NTR}$ in the EGL had any consequence for motor function, floxed *Ngfr* mice were mated with the *Atoh1*-Cre mice to eliminate p75$^{NTR}$ from the EGL while retaining the receptor in Purkinje cells and other neuronal populations. These mice also demonstrated a significant delay in GCP cell cycle exit, a larger cerebellum, and a deficit in motor/balance function on the rotarod test. Although the deficit was not as severe as that seen with the global *Ngfr* knockout mice, these data indicate that the EGL-specific lack of p75$^{NTR}$ during development was sufficient to cause persistent loss of function into adulthood. Thus, the delayed withdrawal from the cell cycle resulting in expanded proliferation of GCPs are likely to have altered the ratio of granule cells to other neuronal populations, impacting the development of appropriate circuitry for motor function.

In summary, we demonstrate a novel function for p75$^{NTR}$ in regulating the timing of cell cycle withdrawal in granule neuron progenitors in the developing cerebellum. ProNT3 specifically antagonized Shh-induced proliferation of GCPs, decreasing the level of HDAC1 and induction of Gli1 mRNA, indicating a potential mechanism for interfering with Shh signaling and facilitating exit of these progenitors from the cell cycle prior to migrating and differentiating. Since precise regulation of these events is critical for normal development, the continued proliferation of GCPs in the

absence of p75$^{NTR}$ led to increased cerebellar size that persisted into adulthood, with deficits in motor behavior.

## Materials and methods

### Primary cerebellum cell cultures

All animal studies were conducted using the National Institutes of Health guidelines for the ethical treatment of animals with approval of the Rutgers Animal Care and Facilities Committee. Cerebella were removed under sterile conditions from P7 pups after euthanizing with $CO_2$. Meninges and small blood vessels were removed under a dissecting microscope. Tissue was minced and dissociated using the papain dissociation kit (Worthington LK003150). Dissociated neurons were plated onto 24 well plates ($1 \times 10^5$ cells in 300 µl of serum free media), 48 well plates ($1 \times 10^5$ cells in 100 µl of serum free media) or 6 well plates ($1 \times 10^6$ cells per well in 1 ml of serum free media) coated with poly-D-lysine (0.1 mg/ml). Serum free medium consisted of 1:1 MEM and F12, with glucose (6 mg/ml), insulin (2.5 mg/ml), putrescine (60 µM), progesterone (20 nM), transferrin (100 µg/ml), selenium (30 nM), penicillin (0.5 U/ml) and streptomycin (0.5 µg/ml). To assay for proNT3 secretion, media was collected from cultures, filtered through 0.22 µm syringe filter, and immunoprecipitated with 2 µg/ml of anti-NT-3 (R&D AF267, RRID:AB_2154250) at 4°C, and probed on a Western blot with anti-proNT-3 (R&D AF3056, RRID:AB_2154250, 1:500).

### Apoptosis assay

Cells were cultured as described above and treated with 2–4 ng/ml of proNT-3 for 48 hr. Cells were then fixed with 4% paraformaldehyde (PFA)/PBS for 15 min and permeabilized with 0.5% Triton $\times$ 100 for 20 min. TUNEL assay was performed following the manufacturers specification (Promega G3250). 10 pictures per coverslip were taken with a Nikon Eclipse TE200 microscope. Number of DAPI and TUNEL-positive cells were quantified using Image J Version 1.49.

### qPCR analysis

Cerebellar neurons were cultured as above for 24 hr in the absence or presence of Shh or Shh +proNT3. RNA was isolated using Trizol (Ambion), and analyzed by quantitative real-time PCR using a Roche 480 II Light Cycler and the Roche Light Cycler sybr green kit. Primers for Gli1 were: forward - GCTGTCGGAAGTCCTATTCAC, reverse - GCCTTCCTGCTCACACATATAA, and for GAPDH: forward - CACCGACCTTCACCATCTTGT, reverse – TTCTTGTGCAGTGCCAGCC.

### Western blot

Tissue or cells were washed with ice-cooled PBS and homogenized using 1% NP40, 1% triton, 10% glycerol in TBS buffer (50 mM Tris, pH 7.6, 150 mM NaCl) with protease inhibitor cocktail (Roche Products, 11 836 153 001). Proteins were quantified using the Bradford assay (Bio-Rad 500–0006) and equal amounts of protein were run on SDS gels and transferred to nitrocellulose membrane. To ensure equal protein levels, blots were stained with Ponceau prior to incubation with antibody. The blots were then rinsed and blocked in 5% nonfat dried skim milk in TBS-T for 2 hr at RT. Blots were incubated with primary antibodies diluted 1:1000 in 1% BSA in TBS buffer overnight at 4°C. The blots were washed with TBS-T 3 $\times$ 15 min each and incubated with Licor secondary antibody for 1 hr at RT. All secondary antibodies were diluted 1:10,000. Membranes were washed 3 $\times$ 15 min each in TBS-T. The membranes were analyzed using Licor Odyssey infrared imaging system (LICOR Bioscience, Lincoln, NE). To confirm equal protein levels, blots were reprobed with actin. All analyses were performed at least three times in independent experiments.

For detection of secreted proNT3, an equal volume of media was immunoprecipitated with anti-NT3 and probed for proNT3 or NT3. For validation of the proNT3 antibody (R&D AF3056, RRID: AB_2154250), equal protein from brain lysates of WT, heterozygous, or *Ntf3*-/- mouse tissue from P0 pups was immunoprecipitated with anti-proNT3 and probed with anti-proNT3 (*Figure 7—figure supplement 1*).

## Immunohistochemistry

Animals were deeply anesthetized with ketamine/xylazine and perfused with 4% PFA/PBS. Brains were removed and postfixed in 4% PFA/PBS overnight at 4°C, then cryopreserved with 30% sucrose. Sections (20 µm) were cut using a Leica cryostat, and mounted onto charged slides. Sections were permeabilized with 0.5% triton in PBS for 10 min and blocked with 1% BSA and 5% donkey serum in PBS for 1 hr at room temperature. Primary and secondary antibodies were prepared in 1% BSA. Sections were incubated with primary antibodies overnight at 4°C in a humidified chamber. Antibodies used were: Ki67 (Abcam 15580, RRID:AB_443209, 1/500), anti-p75 (R&D AF367, RRID:AB_2152638, 1/500), anti-p75 (Millipore MAB365, RRID:AB_2152788, 1/1000), anti-proNT-3 (R&D AF3056, RRID: AB_2154250, 1/200), anti-NT-3 (R&D AF-267-NA, RRID:AB_354434, 1/200), anti-BrdU (Millipore 05–633, RRID:AB_309861, 1/50). All secondary antibodies were diluted 1:1000, and incubated for 1 hr at RT. Nuclei were labeled with 1 µg/ml Dapi/PBS for 1 min at RT or 1 mM Draq5 (BioStatus DR-50200) for 30 min and mounted using Prolong Gold (Life Technologies P36931). Controls for immunostaining included incubation with secondary antibodies in the absence of primary antibodies. The proNT3 antibody was tested for specificity on sections from *Ntf3*-/- mice compared to WT from P0 pups, since the *Ntf3* -/- mice die after birth.

## EdU incorporation assay

For in vivo analysis of proliferation, 40 µl of 10 µM solution of Edu/PBS was injected i.p. 2 hr before perfusing the animals for immunocytochemical analysis. After cryostat sectioning, EdU was developed following the manufacturers protocol (Molecular Probes C10337). Ten pictures per coverslip were obtained on a Zeiss LSM 510meta microscope with LSM acquisition software. Draq5 and EdU positive cells were counted using Image J v1.49.

## Proliferation assays

### In cell western analysis

For in vitro analysis of proliferation, BrdU was added to neuronal cultures at a final concentration of 10 µM at the time of plating with or without 2 µg/ml Shh and the indicated concentration of neurotrophins or proneurotrophins. After 48 hr, cells were washed with ice-cooled PBS and immediately fixed with ice cold 4% PFA/PBS for 10 min. Cells were washed with PBS 2 × 10 min and permeabilized with 0.5% Triton × 100 in PBS for 15 min, and then blocked with 1% BSA, 5% donkey serum in PBS for 1 hr at RT. Cells were incubated with primary antibody overnight at 4°C. Anti-BrdU (Millipore 05–633, RRID:AB_309861) was diluted 1:50 according to the manufacturers specification. The plates were washed with PBS 3 × 15 min each and incubated with Licor secondary antibody diluted 1:5,000 for 1 hr at RT. Cells were then washed 3 × 15 min in PBS. Plates were scanned using Licor Odyssey infrared imaging system (LICOR Bioscience, Lincoln, NE).

### Fluorescence microscopy analysis

Following labeling with anti-BrdU (Millipore BU-1), cells were incubated with a fluorescent secondary antibody (1:1000). 10 pictures per coverslip were taken with a Nikon Eclipse TE200 microscope using MetaMorph acquisition software. The number of DAPI and BrdU-positive cells were quantified using Image J Version 1.49.

The area of cerebellum was measured by tracing the outline of midsagittal sections and analyzing the area using Image J Version 1.49.

## Generation of EGL- specific deletion of p75$^{NTR}$

Mice with homozygous floxed alleles of p75$^{NTR}$ (*Ngfr*$^{fl/fl}$) were mated with *Atoh1*-Cre mice (Jackson Labs), and pups were obtained postnatally at different developmental ages. The genotype of *Ngfr*$^{fl/fl}$; *Atoh1*-Cre animals were confirmed by PCR, and the absence of p75$^{NTR}$ in the EGL was confirmed by immunostaining.

## Behavioral analysis

Balance and coordination were tested using the rotarod assay (*Korpi et al., 1999*). Mice were placed on the rod for training in 5 sessions of 60 s each at low speed (10 rpm), and were tested the following day for 5 sessions at 60 s each with increasing speed in each subsequent session until

40 rpm. After 5 min of rest, the testing protocol was repeated. The number of seconds the mouse stayed on the rod before falling was scored. At least twelve mice per genotype were tested.

## proNT-3 production

HEK 293 cells were grown to 90% confluence in a 10 cm plate and split into 5 plates. The cells were allowed to attach for 4 hr in the incubator. Media was replaced with 5 ml of Optimem (Gibco 31985–070) and cells were returned to the incubator for 20 min. 0.5–1 µg of total DNA and 20 µl of lipofectamine (Thermo Fischer Scientific 18324–0204–020) were mixed in 1 ml Optimem media and left at RT for 20 min. Media was replaced with 4 ml fresh Optimem and the DNA/Lipofectamine was added to the cells overnight. The next day media was replaced with DMEM + penicillin (5 U/ml) and streptomycin (5 µg/ml). After 48 hr the media was collected, aliquoted and stored at −80°C. The concentration of proNT-3 was estimated by Western blot analysis run with a known concentration of NT-3.

## Statistical analysis

Experimental groups were compared using either student's t-test or ANOVA followed by Tukey's posthoc analysis, as appropriate, $p < 0.05$ was considered significant. The specific statistical analysis is indicated in each figure legend.

## Acknowledgements

The authors are grateful to Brian Pierchala for generously providing the floxed p75[NTR] mice, and to Lino Tessarollo for the *Ntf3* knockout tissue. We are also grateful to Alex Joyner for important advice and to Kenneth Teng for helpful discussions. We thank Barbara Hempstead and Augustin Anastasia for providing the plasmid to express uncleaved proNT3, and Aaron Cooper and Subhashani Joshi for excellent laboratory assistance. This work was supported by NIH/NINDS 1R56NS094589 (WJF).

# Additional information

## Funding

| Funder | Grant reference number | Author |
|---|---|---|
| National Institutes of Health | 1R56NS094589 | Wilma J Friedman |

The funders had no role in study design, data collection and interpretation, or the decision to submit the work for publication.

## Author contributions

JPZ, Conception and design, Acquisition of data, Analysis and interpretation of data, Drafting or revising the article; EA, Acquisition of data, Analysis and interpretation of data; WJF, Conception and design, Analysis and interpretation of data, Drafting or revising the article

## Author ORCIDs

Wilma J Friedman, http://orcid.org/0000-0002-3638-3504

## Ethics

Animal experimentation: All animal studies were conducted using the National Institutes of Health guidelines for the ethical treatment of animals with approval of the Rutgers Animal Care and Facilities Committee (protocols 15065 and 15066).

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
