## [Decision Letter]

Thank you for submitting your article "Proneurotrophin-3 Promotes Cell Cycle Exit of Cerebellar Granule Cell Progenitors via the p75 Neurotrophin Receptor" for consideration by *eLife*. Your article has been favorably evaluated by a Senior editor and three reviewers, one of whom, David D Ginty (reviewer #1), is a member of our Board of Reviewing Editors.

The reviewers have discussed the reviews with one another and the Reviewing Editor has drafted this decision to help you prepare a revised submission.

Summary:

This manuscript addresses the role of p75^NTR^ in cerebellar development. The authors convincingly demonstrate that mutations in p75, either germ-line or restricted to the *Atoh1*- expressing granule cell lineage, cause persistent proliferation and delayed cell cycle exit of GCPs. Surprisingly only pro-NT3 and not the other pro-neurotrophins can stimulate p75 and regulate cell cycle exit in GCPs ex vivo. The data indicate that proNT3 functions to oppose Shh-induced proliferation. Overall the data are clear and convincing. The findings shed light on the functions of p75 and cerebellar development.

Essential revisions:

1) A major point of the paper is that proNT3 expressed in Purkinje cells is the relevant p75 ligand. Others have failed to detect NT3 mRNA in Purkinje cells (Lindholm et al., 1993). The immunostaining and Western blot data presented in Figure 7 are not validated with the necessary controls with cells and tissues lacking NT3. ProNT3 immunostaining and immunoblots should be done using NT3 mutant mice to evaluate antibody specificity.

Related to this, if NT3 animals are available to the authors at the appropriate developmental age, then it would be desirable to examine whether GCPs in these mice exhibit persistent proliferation and delayed cell cycle exit.

2) Additional experiments are needed to substantiate the SHH argument. Measuring Gli1 and/or Ptch1 levels of expression should be possible by RNA or protein (for Glib).

3) The role of p75 in cell cycle regulation in the developing CNS has been previously investigated. See in particular: Neuronal cell cycle: the neuron itself and its circumstances José M Frade & María C Ovejero-Benito (2015) Cell Cycle, 14:5, 712-720 and references therein. The Abstract and Introduction should be changed to reflect these prior findings.

---

## [Author Response]

Essential revisions:

1) A major point of the paper is that proNT3 expressed in Purkinje cells is the relevant p75 ligand. Others have failed to detect NT3 mRNA in Purkinje cells (Lindholm et al., 1993). The immunostaining and Western blot data presented in Figure 7 are not validated with the necessary controls with cells and tissues lacking NT3. ProNT3 immunostaining and immunoblots should be done using NT3 mutant mice to evaluate antibody specificity.

It is true that NT3 mRNA has only been detected in granule cells, however several previous publications have shown NT3 protein in Purkinje cells and not granule cells (Zhou and Rush, 1994; Friedman et al., 1998), each study using different antibodies. Zhou and Rush even commented on the fact that there was a mismatch with the mRNA being seen in granule cells and the protein in Purkinje cells. It is difficult to explain why the mRNA has been detected in granule cells and the protein in Purkinje cells, however we have now validated the antibodies used here by immunoprecipitation of NT3+/+, NT3+/- and NT3-/- mouse tissue, generously provided by Lino Tessarollo, using the anti-proNT3 antibody. Bands were seen only in the NT3+/+ and NT3+/- tissue, but not in the NT3-/- tissue, and the band was the same size as the proNT3 we produced from HEK cells for these studies. NT3 is expressed at very low levels in the developing brain, and since the NT3-/- mice die at birth we were only able to obtain P0 brains. Immunostaining was detected in the cingulate cortex of WT brains, as seen for NT3 mRNA by in situ hybridization (Friedman et al., J, Neurosci. 11:1577-1584, 1991) and NT3-LacZ reporter expression (Vigers et al., J. comp. Neurol. 416:398-415, 2000), but not NT3-/- brains. Additionally, the anti-proNT3 antibody was able to reverse the effect of proNT3 on GCP proliferation in vitro (Figure 5). Therefore, we are confident that the proNT3 antibody specifically recognized proNT3. Future studies to resolve the source of the proNT3 to granule or Purkinje cells may be done with floxed NT3 mice crossed with the *Atoh1*-Cre, or a Purkinje cell Cre, but that is beyond the scope of the current study.

Related to this, if NT3 animals are available to the authors at the appropriate developmental age, then it would be desirable to examine whether GCPs in these mice exhibit persistent proliferation and delayed cell cycle exit.

We agree that it would be very helpful if we could examine NT3-/- mice at these ages to determine whether they phenocopy the p75 null animals, however the NT3 null mice die at birth (Tessarollo et al. PNAS 94:14776-14781, 1997).

2) Additional experiments are needed to substantiate the SHH argument. Measuring Gli1 and/or Ptch1 levels of expression should be possible by RNA or protein (for Glib).

We have now investigated whether induction of Gli1 mRNA by Shh is prevented by proNT3, and we now include data demonstrating that proNT3 reduced Shh induction of Gli1 mRNA, supporting our hypothesis that regulation of HDAC1 by proNT3 may prevent or attenuate the ability of *Gli2* to regulate downstream transcriptional events. Those data are now included in Figure 8.

3) The role of p75 in cell cycle regulation in the developing CNS has been previously investigated. See in particular: Neuronal cell cycle: the neuron itself and its circumstances José M Frade & María C Ovejero-Benito (2015) Cell Cycle, 14:5, 712-720 and references therein. The Abstract and Introduction should be changed to reflect these prior findings.

These references have now been included in the Introduction.